# The Use of Isoflurane and Adjunctive Magnesium Chloride Provides Fast, Effective Anaesthetization of *Octopus vulgaris*

**DOI:** 10.3390/ani13223579

**Published:** 2023-11-20

**Authors:** Anna Di Cosmo, Valeria Maselli, Emanuela Cirillo, Mariangela Norcia, Heethaka K. S. de Zoysa, Gianluca Polese, William Winlow

**Affiliations:** 1Department of Biology, University of Naples Federico II, 80126 Naples, Italy; valeria.maselli@unina.it (V.M.); emanuela.cirillo@unina.it (E.C.); mariangela.norcia@unina.it (M.N.); krishanthasameeradezoysa.heethaka@unina.it (H.K.S.d.Z.); gianluca.polese@unina.it (G.P.); 2PNRR “MNESYS”, University of Naples Federico II, 80126 Naples, Italy; 3Department of Bioprocess Technology, Faculty of Technology, Rajarata University of Sri Lanka, Mihintale 50300, Sri Lanka; 4Institute of Ageing and Chronic Diseases, University of Liverpool, Liverpool L69 3BX, UK

**Keywords:** general anaesthetics, *Octopus vulgaris*, isoflurane, magnesium chloride, depth of anaesthesia, stress reduction

## Abstract

**Simple Summary:**

Anaesthetising invertebrates is a welfare issue, but because there are so many different invertebrates (97% of all animal species), it is of the utmost importance to work out the most appropriate way to anaesthetise each group or even particular species of animals. In order to reduce stress and pain to the animals, clinical anaesthetics offer a standardised approach, given their known and well-studied effects at both system and cellular levels. Here, we show that the clinical anaesthetic isoflurane (1%), often used by veterinarians, can anaesthetise octopuses quickly and efficiently when used in conjunction with 1% magnesium chloride, which acts as a muscle relaxant. Recovery is equally quick, and the whole process from anaesthetisation to recovery is completed in 30 to 35 min, minimising animal stress.

**Abstract:**

A wide variety of substances have been used to anaesthetise invertebrates, but many are not anaesthetics and merely incapacitate animals rather than preventing pain. In essence, the role of an ideal general anaesthetic is to act as a muscle relaxant, an analgesic, an anaesthetic, and an amnesic. To achieve all these properties with a single substance is difficult, and various adjuvants usually need to be administered, resulting in a cocktail of drugs. In a clinical setting, the vast majority of patients are unaware of surgery being carried out and have no memory of it, so they can claim to have felt no pain, but this is much more difficult to demonstrate in invertebrates. Here, we show that 1% MgCl_2,_ a muscle relaxant_,_ is a useful adjuvant for the clinical anaesthetic isoflurane on *Octopus vulgaris* when applied alone in seawater for 10 min before the clinical anaesthetic. After this, full anaesthesia can be achieved in 5 min using 1% isoflurane insufflated into the saline still containing MgCl_2_. Full recovery takes place rapidly in about 10 to 15 min. The depth of anaesthesia was monitored using changes in respiratory rate, chromatophore pattern, and withdrawal movements of the arms and siphon. This methodology reduces stress on the animal and minimises the quantity of anaesthetic used.

## 1. Introduction

In the past, the welfare of invertebrates was largely unexplored, and little attention has been paid to it by many experimental biologists. According to Mather [1], “Consideration of welfare of other animals often is anthropocentric, focussing on mammals similar to humans”. Fortunately, this view is gradually changing as it becomes clear that other groups of animals may be sentient and capable of feeling pain and suffering in consequence, particularly decapod crustaceans [2,3], cephalopod molluscs [4,5,6], and possibly some insects. This may also be true of other groups of animals, and it behoves us to treat all invertebrates with respect when carrying out experimental procedures upon them.

Many different substances have been suggested and used for anaesthetising invertebrates [7], some more successfully than others. Clinically used general anaesthetics have several functions, and many different types exist, either volatile or systemic anaesthetics. We know the general principles behind their use because of the vast amount of clinical experience with them. In essence, the role of an ideal general anaesthetic is to act as a muscle relaxant, an analgesic, an anaesthetic, and an amnesic. Achieving all these properties is difficult with a single substance, and various adjuvants usually need to be used, resulting in the administration of a cocktail of drugs. In a clinical setting, the vast majority of patients are unaware of surgery being carried out and have no memory of it, so they can claim to have felt no pain, but this is much more difficult to demonstrate in invertebrates. However, there is strong support for the view that cephalopods can feel pain [8,9,10]. For example, they avoid locations previously associated with noxious stimuli [11]. There is also accumulating evidence for pain in arthropods [12]. Thus, we need to exert great care when handling all invertebrates to reduce stress and pain to the animals, particularly when carrying out physiological experiments.

For some time, it has been suggested that magnesium chloride by itself can act as an anaesthetic agent in cephalopods (e.g., [13,14,15]), but its main use is as a muscle relaxant, which may render an animal immobile. However, we cannot ensure that the animal is pain-free, and there has been substantial controversy on this issue [7,16]. Magnesium chloride works by competing with calcium ions to prevent synaptic release of neurotransmitters in the periphery and does not usually gain access to the central nervous system [5]. In 2018, we suggested that MgCl_2_ might be useful as an adjunct to anaesthesia [5], and here we explore this concept in more detail. There is a fuller discussion of the mode of action of magnesium salts as muscle relaxants elsewhere [5], but it should be noted that MgCl_2_ may also be a mild central sedative analgesic [17,18]. However, in our view, volatile anaesthetics are preferable, and isoflurane has proved successful for anaesthetising *Octopus vulgaris* [5,19]. Isoflurane is a non-flammable halogenated ether compound used by both clinicians and veterinarians. When administered via the respiratory system, it has direct access to both the peripheral and central nervous systems.

***Properties of general anaesthetics.*** Clinical anaesthetics, such as isoflurane and related compounds, have well-known systemic and cellular actions, which have been demonstrated on mammals and on gastropod molluscs such as *Lymnaea stagnalis* [19] and on *Octopus vulgaris* [20]. They include ion channels and voltage-gated channels for sodium, potassium, and calcium and tend to block synaptic transmission [21]. These are general properties and actions and should not be ignored by investigators working on invertebrates. Furthermore, Keltz and Mashhour [22] recently provided good evidence that a generalisable mechanistic framework for the actions of general anaesthetics is emerging from studies of a wide range of species “from *Paramecium* to primates” and includes *Drosophila*, *Caenorhabditis elegans*, gastropod molluscs, such as *Lymnaea stagnalis*, etc., as well as plants.

We recently developed a successful method for anaesthetising *Octopus vulgaris* using the clinical anaesthetic isoflurane [20] without any adjunctive agents, but this took approaching two hours from the start of anaesthetisation to full recovery. In these experiments, we gradually increased the concentration of isoflurane in seawater from 0.5% to 2.0% and found that at 1.0% isoflurane, the chromatophores started to flash in an uncontrolled manner, a sign of loss of central motor control of chromatophores in *Octopus vulgaris* [23]. After this, the animals became relaxed, unresponsive to touch stimuli, and anaesthetised [20]. Recovery and anaesthetisation were judged physiologically by the rate of respiratory pumping because most clinical anaesthetics depress minute ventilation in mammals [24]. Two behavioural tests were also used: local withdrawal responses of the arms and siphon and loss of chromatophore patterning. A shorter period of anaesthetisation should reduce both the stress on the animal and the quantity of anaesthetic used. Here, we report, for the first time in *Octopus vulgaris*, that the use of MgCl_2_, as both a pre-anaesthetic agent and as an adjunct to anaesthesia, significantly reduces the time needed to anaesthetise the animal successfully and also promotes more rapid recovery.

## 2. Materials and Methods

### Animals

Nine specimens of *Octopus vulgaris* (5 males and 4 females; body weight, 0.5–1.0 kg) were captured in the Bay of Naples, bought from the local fish market, and transported to Prof. Di Cosmo’s cephalopod facility [25] at the Department of Biology, University of Naples Federico II (Naples, Italy). Our research was approved following the European Directive 2010/63 EU L276, the Italian DL. 4/03/2014, n°26, and the ethical principles of Reduction, Refinement, and Replacement (Project n°608/2016-PR-17/06/2016; protocol n°DGSAF 0022292-P-03/10/2017). The test specimens were acclimatised [26], and were fed with fresh fish (*Engraulis encrasicolus*) as described in Maselli [27]. All octopuses fully recovered from the experimental procedures; they were kept in Prof. Di Cosmo’s animal facility and later employed in other experiments.

*Anaesthesia setup*—In our study, inhalational anaesthesia was performed as described by Polese et al. [20]. The seawater surrounding the animal was aerated using a power pump and regulated by a flowmeter (1.8 L/min) in series with an isoflurane vapouriser and delivered to the bath via an air stone as previously described [20]. Except for the delivery of the anaesthetic into the aquatic medium, this is a standard clinical technique that allows the isoflurane to be delivered to the animal with air as the carrier gas, thus ensuring that the animal remains fully oxygenated throughout the procedure. All inhalational anaesthetics are delivered into the animal’s system via an air–water interface, the respiratory epithelium in both vertebrates and octopuses, but in this case, it was via an aquatic medium. The experimental tank was kept in an enclosure and vented externally. It contained 1600 mL of seawater and was kept at 18 °C throughout the experiments. Each animal was left to acclimatise in the experimental tank for 10 min before delivering the air and isoflurane gas mixture via the air stone.

***Protocol for anaesthesia***—After the acclimatisation period, the animals were first moved gently into an experimental tank containing 1% MgCl_2_ in aerated seawater for a period of 10 min. Then, the anaesthetic was insufflated in the airflow for 5 min at 1% concentration, after which the animal was maintained in fresh, clean seawater until full recovery. To determine the depth of anaesthesia, we monitored the respiratory rate as judged by the frequency of respiratory pumping and behavioural changes in the chromatophore pattern and withdrawal of the arms and siphon. We recorded the responses with a web camera (GoPro Hero10, 4K video at 30 frames per second (fps) in full-screen; internal auto-focus lens system).

The respiratory rate was taken every minute just before the touch test was performed, to avoid any change due to stimulation. The animals had widely varying pre-control respiratory rates (24–45 cycles per minute) as judged by the pumping movements of the mantle; so, for analytical purposes, we normalised the respiratory rate, with 100% being the observed pre-control value [20].

Touch stimuli induced withdrawal of the arms and siphon, and we monitored the changes in chromatophore pattern throughout each experiment. The withdrawal responses were elicited by the gentle application of a blunt Plexiglas probe to the arms and siphon. For analytical purposes, we categorised the responses as strong, medium, weak, or abolished, and these were given numerical values of 6, 4, 2, and 0, respectively. Using ImageJ software 1.46r [28], we measured a spot 10 pixels in size from the interbrachial membrane on frames taken every 5 min throughout the experiments. The colour intensity of each spot was measured based on the additive red, green, blue (RGB) colour model whereby a zero intensity (value, 0) for each component gives the darkest colour (no light, considered to be black) and full intensity for each component gives white (value, 255).

*Statistical analyses*—A one-way analysis of variance (ANOVA) was used to assess the significance of differences between the data at time 0 and those at other times (R-CRAN, [29].

## 3. Results

### 3.1. Actions of 1% MgCl_2_ on Animal Behaviour

Upon entry to the 1% MgCl_2_ in aerated seawater, most animals struggled, but within three minutes of application of MgCl_2,_ the animals visibly started to relax (4.17% reduction in the median normalised respiratory rate), and by 5 min, all were pale in colour (Figure 1) with weak touch reflexes (Figure 2). After 10 min, the animals were all pale, unreactive, and immobile, with a 19.50% reduction in the median % normalised respiratory rate. Thus, the animals appeared to be paralysed but continued to breathe steadily (Figure 3).

None showed flashing reactions due to chromatophore activation at any stage, presumably due to the inactivation of transmitter release from chromatophore motor neurons.

### 3.2. Addition of I% Isoflurane

After 10 min in 1% MgCl_2_, 1% isoflurane was bubbled into the experimental tank. The animals remained unreactive and became paler (Figure 1), but respiration continued at a lower rate as the isoflurane took effect (Figure 3). Reduced respiratory rate is a clear sign that anaesthetics are taking effect in clinical and veterinary settings. After 5 min of anaesthetisation, the isoflurane was switched off, and the normalised respiratory rate was reduced to 54.17% ± 3.59% (median ± se) of the normalised value. At this time, the pale and unresponsive animals were gently moved to a different experimental tank containing fresh aerated seawater to recover.

### 3.3. Recovery from Anaesthesia

In 6 cases, the animals achieved full recovery of colour pattern, respiratory frequency, reflex arm, and siphon sensitivity in 10–15 min. One animal recovered quickly in 7 min and two others in about 20 min.

Both parameters, colour intensity of the interbrachial membrane and normalised respiratory rate, showed statistically significant differences after 10 min, 15 min and 25 min, compared with data at time 0 (ANOVA one way, *p* < 0.05).

## 4. Discussion

In the experiments described here, we demonstrated that *Octopus vulgaris* can be pre-anaesthetised with MgCl_2_ for 10 min and then fully anaesthetised with 1% isoflurane for five minutes with a maximum recovery period of about 20 min.

We observed that the normalised respiratory rate (%) decreased gradually during the first 10 min of the experiment due to the presence of the MgCl_2_ alone, declining by a normalised value of 20%. Subsequently, adding isoflurane (1%) induced a significant decrease in the respiratory rate, declining by 44%. This effect carried over into the subsequent phase in which the animal was put in fresh aerated seawater but started to decline after a further 2 min, with considerable variation between animals. However, after 20 min (35 min into the experiment overall) of refreshing the bathing solution with clean aerated seawater, all animals recovered, but the normalised respiratory rate did not reach 100% in all animals (median ± standard error, 90.69 ± 1.60). However, all the animals were in good condition. The animals were monitored for the following 7 days, and all were healthy and behaving normally.

This compares very well with our earlier study [20], where increasing doses of isoflurane (0.5 to 2.0% in seawater) were gradually applied to 10 specimens of *Octopus vulgaris* over a period of about 40 min, followed by a recovery period of up to 1 h. Clearly, this new protocol is much less stressful for the animals and utilises much less isoflurane. In addition, the animals showed few signs of discomfort in MgCl_2,_ and there were no flashing responses as the animals became paler in colour and touch reflexes declined as the animals became immobilised. However, in 1% MgCl_2_ in aerated seawater, the respiratory rate declined by less than 20% and stabilised after about 10 min. Upon the addition of 1% isoflurane, the respiratory rate gradually declined as expected since a declining respiratory indicates an increasing depth of anaesthesia. Furthermore, isoflurane is known to be suitable for maintenance anaesthesia [30] as it also enhances muscle relaxation, while MgCl_2_ diminishes the early excitatory phase of anaesthesia [31], as we previously suggested [5].

Although magnesium chloride has been used as an anaesthetic for octopods and other cephalopods [16], it is usually applied externally, giving little or no access to the CNS. It is our view that it is wholly inappropriate as an anaesthetic because MgCl_2_ is a basic muscle relaxant, although it may have minimally sedative properties as mentioned above [20]. We therefore refute the suggestion that it can be used as an anaesthetic substance in its own right as suggested elsewhere [16], where increasing concentrations of MgCl_2_, up to 3.75%, were used to paralyse specimens of octopus and cuttlefish over about 20 min and were compared with the effects of ethyl alcohol, another non-anaesthetic. Apparently, these substances reversibly depressed evoked activity in the pallial nerve, but no records were shown. In order to clarify nervous activity under these circumstances, recordings of electrical activity need to be clearly demonstrated, perhaps using the recently published methods of Gutnick et al. [21].

Appropriate maintenance anaesthesia for experimentation has not yet been determined for *Octopus vulgaris,* and further experiments are required to determine the most appropriate level of anaesthesia in fresh animals relaxed with MgCl_2_, which will probably require a slightly higher concentration of isoflurane if the animal is to undergo surgery.

## 5. Conclusions

1% magnesium chloride is a useful pre-anaesthetic agent for the isoflurane anaesthetisation of *Octopus vulgaris*. It reduces the concentration of isoflurane necessary for complete anaesthetisation and significantly reduces the stress on the animal and the time course of anaesthesia.

*Note on the use of vaporisers*. The application of volatile general anaesthetics need not involve the use of vaporisers, although they simplify the protocol. Instead, volatile anaesthetics can simply be dissolved in saline at appropriate concentrations, as performed by Dickinson [22]. Isoflurane is usually a controlled substance, not freely available, and is best obtained via collaborating veterinarians.

## Figures and Tables

**Figure 1 animals-13-03579-f001:**
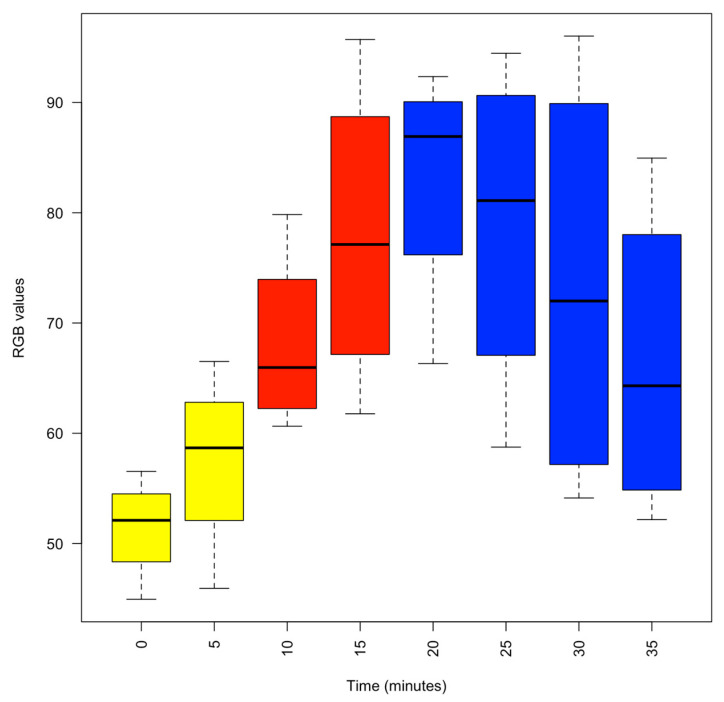
Box plot of the colour intensity of the interbrachial membrane as the concentration of isoflurane was progressively increased and then decreased. Yellow, 1% MgCl_2_; Red, 1% MgCl_2_ and 1% isoflurane; Blue, recovery in fresh aerated seawater. Colour intensity was measured using the RGB model whereby a zero intensity (value, 0) for each component gives the darkest colour (no light, considered to be black) and full intensity for each component gives white (value, 255). Thus, the higher the recorded value, the paler the interbrachial membrane and vice versa.

**Figure 2 animals-13-03579-f002:**
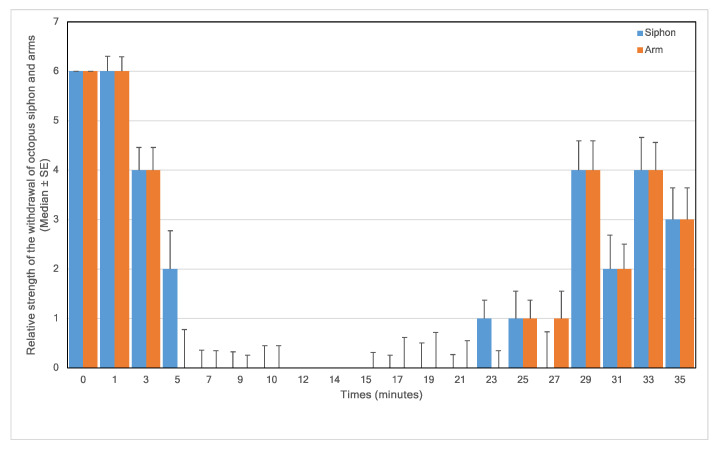
Relative strength of the withdrawal of octopus siphon and arms in response to a touch stimulus versus time (6 = strong, 4 = medium, 2 = low, and 0 = none) in 9 animals; 0–10 min in 1% MgCl_2_; 10–15 min in 1% MgCl_2_ and 1% isoflurane; 15–35 min in fresh aerated seawater.

**Figure 3 animals-13-03579-f003:**
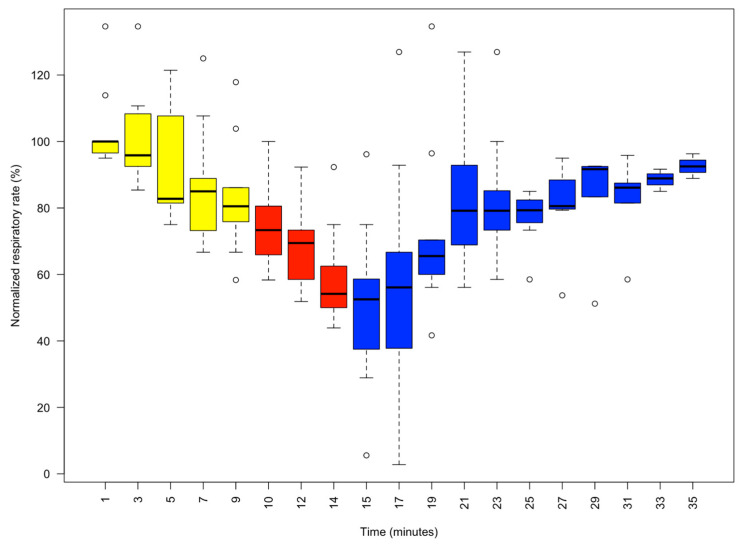
Boxplot of the normalised respiratory rate (%) as determined by the number of mantle contractions per minute in 9 animals. Readings were taken at 2 min intervals. Yellow phase: MgCl_2_ 1% (10 min). Red phase: isoflurane 1% and 1% MgCl_2_ (5 min). Blue phase: recovery phase in fresh aerated seawater. As respiratory rate variability between animals is substantial, the outlying circles indicate the recorded maximum and minimum normalised respiratory rates. It should be noted that after 10 min in 1% MgCl_2_, the respiratory rate stabilised at about 80% of its normalised value, followed by a further gradual decline after 1% isoflurane, which was terminated by changing the medium to clean aerated seawater.

## Data Availability

Data are contained within the article.

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
