# Peer review of "The Use of Isoflurane and Adjunctive Magnesium Chloride Provides Fast, Effective Anaesthetization of Octopus vulgaris"

_animals, 2023, doi:10.3390/ani13223579_

Round 1
Reviewer 1 Report
Comments and Suggestions for Authors
This study aims to find methods for effective anaesthesia of Octopus and reports a protocol that appears to advance that aim. The authors had previously reported on the use of isoflurane as an anaesthetic (2014) but in the present study examines pretreatment with magnesium chloride as an adjuvant that appears to offer advantages for the welfare of the subjects. There are some issues with the manuscript, some related to the structure and language but some relate to the analysis. These are noted by line number.
2 The title is broad and might better suit the ms if it focused on cephalopods or Octopus. For example, deleting “appropriate anaesthetics for invertebrates” would give a more accurate title.
38 I suggest a comma after pain
51 This does not follow I suggest deleting “but this is much more difficult to demonstrate in invertebrates”.
53 However is not appropriate because this is a different topic. It is a topic from the first paragraph, which makes a case for pain in invertebrates. Put the required additional information on pain in paragraph one and delete lines 53-57. Then insert a sentence or two which makes the case for good anaesthesia for octopods so that their welfare can be improved.
60 ….animal immobile. However, we cannot be sure that the animal is pain free…
61 I suggest replacing “It” with “magnesium chloride”.
67 There is a problem here. Should it be However, in our view, volatile…
72 The sentence does not make sense.
77 comma after “after this”
83 I suggest …..reduce both the stress on the animal and the quantity of….
89 I would start this sentence with “Nine” rather than the number but this might be the journal style.
95 Give some information on how the subjects were acclimatized so we do not need to look at other studies.
116 I suggest deleting “used two criteria: physiological” and replace with “monitored”
135 I presume the ANOVA was a repeated measures ANOVA.
Results : I cannot find any reference to the results of the ANOVA tests? If they were conducted, they should be reported. At the moment, the results are rather descriptive and focus on particular phases of the protocol rather than overall changes. We should be told if these changes are significant.
146-7 The section starting with “presumably” is discussion rather than results.
194 Problem with sentence.
198 “Cleary this new protocol is much less stressful for the animals….. “ we need more information. How was stress assessed in the study? This is a key part of the paper because it is intended for a collection of papers on welfare. Are there other methods of assessing stress that might be used in future studies?
208 octopods
230 lower case for M and A.
Comments on the Quality of English Language.
Author Response
Comments to Reviewer 1 Thank you for carefully reviewing this manuscript. We have made the appropriate changes as you suggested, except for the original lines 146-7 which we have left in place as we believe them to be an explanation at the most appropriate point in the text.Reviewer 2 Report
Comments and Suggestions for Authors
The manuscript describes the use of isoflurane in conjunction with MgCl2 for the anesthetization of octopus to improve animal welfare during experiments. The experiment is generally well-planned. The methodology and the result are clearly described. The knowledge from this study can be used as a guideline for appropriate anesthetization of octopus in laboratory conditions. However, the content of the manuscript is too short to be published as an “Article”. Only one fixed dose of isoflurane (1%) and MgCl2 (1%) combination and one treatment duration (i.e., 10 min-bath with MgCl2 followed by 5 min-bath with isoflurane + MgCl2) was tested. The assessment of the depth of anesthesia was based on only 3 parameters which were respiratory rate, changes of the chromatophore pattern, and withdrawal of the arms and siphon. Therefore, this manuscript is more suitable to be published as a “Short Communication” instead of an “Article”.
Other comments are as follows:
1. The title should be more specific. There is only one anesthetic and one invertebrate animal in this study. Thus, the word “anesthetics” and “invertebrate” should be replaced by “isoflurane” and “octopus (or Octopus vulgaris)”, respectively.
2. In the abstract, the introduction part (Line 20-28) is too long. The methodology is absent altogether and the description of results (Line 28-31) is too short.
3. For statistical analysis, the statistical software should be mentioned in the Materials and Methods. Also, the results of statistical analysis are absent from the manuscript.
4. The authors stated that one-way ANOVA was used to assess the statistical difference (Line 136). However, the data of “the relative strength of the withdrawal siphon and arms in response to a touch stimulus” which is categorized as strong, medium, weak, or abolished (Line 129-130 and 157) is on an ordinal scale. It should be noted that one-way ANOVA cannot be used for analyzing the data in ordinal scale. In this case, a non-parametric Kruskal-Wallis should be applied.
5. According to the methodology (Line 113-115 and 157-158), the authors administered MgCl2 alone at 0-10 min, isoflurane + MgCl2 at 10-15 min, and move the octopus into the recovery tank at 15-35 min. Thus, the red box in Figure 1 (indicating isoflurane + MgCl2) at 20 min should be replaced by a blue box (indicating the recovery phase).
6. The y-axis of Figure 2 (“mean ± SE”) should be changed to “Relative strength of the withdrawal of octopus siphon and arms”
7. Line 220-229 “Properties of general anaesthetics....” is not a discussion. It should be moved to the Introduction part.
8. The conclusion of “1% magnesium chloride is a useful pre-anesthetic agent for isoflurane anaesthetisation of Octopus vulgaris. It reduces the concentration of isoflurane necessary for complete anaesthetization, significantly reduces the stress on the animal and the time course of anaesthesia.” (Line 234-237) is not supported by the experiment because the authors did not compare the effects of isoflurane with and without MgCl2 (i.e., no control group).
Author Response
Comments to Reviewer 2 Thanks for your comments, but we do not agree that this should be a short publication as it is the culmination of our studies first published in 2014 (Polese et al). The current apear was seriously delayed by the COVID-19 crisis. We have modified the title as suggested and also modified the abstract as suggested. We have also given further information on the statistical package used. We have modified Figure 1 as you suggested and also changed the y-axis on Figure 2. Original lines 220-229 now appear in the introduction. As to your point 8, the previous 2014 paper in which no MgCl2 was used effectively provides the control group for the present work.